# Sleep Disturbances and Dementia in the UK South Asian Community: A Qualitative Study to Inform Future Adaptation of the DREAMS-START Intervention

**DOI:** 10.3390/geriatrics10050121

**Published:** 2025-09-08

**Authors:** Penny Rapaport, Malvika Muralidhar, Sarah Amador, Naaheed Mukadam, Ankita Bhojwani, Charles Beeson, Gill Livingston

**Affiliations:** 1UCL Division of Psychiatry, 6th Floor Maple House, 149 Tottenham Court Road, London W1T 7NF, UK; m.muralidhar@qmul.ac.uk (M.M.); s.amador@ucl.ac.uk (S.A.); n.mukadam@ucl.ac.uk (N.M.); a.bhojwani@ucl.ac.uk (A.B.); charles.beeson.22@ucl.ac.uk (C.B.); g.livingston@ucl.ac.uk (G.L.); 2North London Partnership NHS Foundation Trust, London NW1 0PE, UK

**Keywords:** dementia, sleep disturbance, ethnicity, South Asian, psychological intervention, cultural adaptation

## Abstract

**Background/Objectives**: Little is known about experiences of sleep disturbance in dementia amongst South Asian families, the UK’s biggest minority ethnic group. We aimed to explore their experiences of these alongside translation and preliminary cultural adaptation of an existing effective multicomponent intervention, DREAMS-START. **Methods**: We interviewed family carers of people living with dementia who had participated in the DREAMS-START (*n* = 7) trial or other dementia studies (*n* = 4), conducting reflective thematic analysis. We translated DREAMS-START into Hindi and consulted with Hindi speakers with experience in dementia care, revising iteratively. **Results**: We identified two overarching themes: (i) the experience of dementia-related sleep disturbance in South Asian families, including the impact of multigenerational living, cultural expectations and practices, and existing relationships; and (ii) considerations for culturally adapting DREAMS-START, including language barriers, linguistic and other changes to peripheral elements to increase engagement and relevance, and culturally competent facilitation of the intervention. **Conclusions**: Consideration of multigenerational family structures, within-member dynamics, culturally appropriate activities and lack of access to support are important during consultation and intervention. It was thought that cultural adaptation of the intervention in language and facilitator cultural competence, including consideration of the schedule of prayer, would increase relevance and thus community access. We will use this preliminary work to inform future cultural adaptation and testing of the intervention with the intention to widen access for UK-based South Asian families.

## 1. Introduction

Sleep disturbances are common in people with dementia [1,2,3] who are mostly cared for at home by family members [4]. Relatives are also impacted by sleep disturbances, which may lead to a breakdown of care at home and care home transition [5,6,7,8]. Non-pharmacological interventions should be first-line management, avoiding pharmacological side effects. Multi-component interventions show promise, but until recently, there were no full trials for effectiveness [9,10,11]. DREAMS-START (Dementia related manual for sleep: Strategies for relatives), a six-session manual-based multi-component intervention, is clinically effective with significant improvements in both person with dementia and carer sleep after eight months [12]. This personalised intervention is delivered by trained but not clinically qualified facilitators to family carers, who then implement strategies supporting their relatives to make changes to their sleep. The intervention includes psychoeducation on sleep and dementia, light therapy and activity, supports carers to use practical strategies on sleep hygiene, day and nighttime routine, and helps carers look after their own sleep [13]. Although work continues to develop evidence-based interventions targeting sleep disturbances for people living with dementia, a disproportionate number of dementia research participants are from White, socially advantaged populations [14,15,16,17].

UK-based South Asian people (the largest non-white group) are younger at both diagnosis and death with dementia than White people [18]. Sleep disturbance has been explored in Black, Hispanic, and non-Hispanic White populations [19,20] but little is known about the experiences of sleep disturbance and its wider impact in South Asian families. There may be important cultural differences between populations [21] which should be considered when designing interventions. South Asian families often live in multigenerational households, share caring responsibilities, and may be particularly reluctant to use care homes [22]. Consequently, more relatives may be impacted by sleep disturbances. In addition, such groups experience barriers to help-seeking, leading to later presentation to services [23]. There is a need for interventions tailored to meet the needs of the UK-based South Asian people to reduce disparity and ensure appropriate care pathways and interventions [18]. Services that are not culturally targeted and appropriate, may lead to poorer dementia outcomes in minority ethnic groups [24,25]. As we now have the intervention DREAMS-START, which has potential for delivery at scale in the UK health service, we want to ensure that the intervention will also be accessible and acceptable for minoritized communities.

In this paper, we report on qualitative work within the DREAMS-START trial aimed at understanding how sleep disturbances are experienced by UK-based South Asian families affected by dementia. We discuss how findings may inform cultural adaptation of DREAMS-START, training for facilitators, and describe our process of translating the intervention into Hindi, a common South Asian language, ready for testing in future research.

## 2. Methods

### 2.1. Ethics

The trial is registered ISRCTN 13072268 and was approved by London—Camden and Kings Cross Ethics Committee (20/LO/0894) on 21 August 2020, with an amendment approved to conduct this sub-study on 7 July 2022. All participants gave informed consent.

### 2.2. Procedures

#### 2.2.1. Participants

We invited South Asian family carers of people with dementia and sleep disturbance to participate. Most participated in the intervention arm of the DREAMS-START RCT and were recruited following their trial follow-up assessment. We also included South Asian participants from previous dementia studies who had agreed to further contact [26,27]. To achieve maximum variation, we purposively recruited carers with varying demographics backgrounds (sex, age, South Asian sub-group, level of education, employment status), relationships to the person with dementia, and caregiving contexts (living with relative with dementia or not, relative at home/relative in care home/ex-carers). We included completers and non-completers of DREAMS-START.

#### 2.2.2. Qualitative Data Collection

MM, a Hindi-speaker, interviewed family carers in person or remotely based on participant preference. Carers were offered a professional interpreter if their first language was not English. We developed a semi-structured topic guide based on previous work [27] (see Appendix A for topic guide). We included prompts regarding family carers perception of being a ‘carer’, why it was the specific family member who became the carer for their relative with dementia, sharing information with other family members, and perspectives on linguistic translation and facilitator background. Interviews were audio recorded, translated where necessary, transcribed verbatim, anonymized and entered into NVivo 12 software.

### 2.3. Data Analysis

We undertook a reflexive thematic analysis, identifying themes and patterns within the data [28]. MM conducted the initial coding. All transcripts were open coded. Each interview was coded immediately after transcription, to adapt the topic guide and add prompts exploring new concepts from the interviews. MM listened to all audio recordings, read and re-read the transcripts to identify initial codes. Once all the interviews were open coded, the wider team (GL, PR, NM, and SA) met to review and discuss codes and agree final themes, refining them iteratively based on discussion. We ceased interviews at thematic saturation, at the point that the researcher coding an interview identified no new codes and when the authors’ reflections on additional interviews resulted in no further emergent themes. Our analysis was informed by the Cultural Treatment Adaptation Framework [29] (see Figure 1). The framework defines concepts and language for adapted treatment components, allows differentiation between the core therapeutic components and peripheral components which include how the core components are delivered so they are more understandable and culturally appropriate to improve the ability of the intervention to reach and involve participants.

### 2.4. Researcher Reflexivity

MM is Indian, living in England, with a psychology background, dementia research experience, personal experience of relatives living with dementia, and an understanding of how mental health and dementia are perceived in a South Asian community. She maintained a reflexivity diary throughout data collection and analysis to consider her interpretation of ideas and individual characteristics that may have impacted responses, which were discussed with the research team [28].

### 2.5. Translating DREAMS-START Manuals into Hindi

In parallel to our qualitative work, we translated the intervention into Hindi. Whilst Hindi is less commonly spoken in the UK than other South Asian languages such as Urdu, Punjabi and Bengali, we made a pragmatic decision within the limitations of this small-scale study to translate into Hindi as MM and AB were native Hindi speakers and were able to lead on the translation process. We consulted with leaders in a South Asian community centre and MM presented to a group of older people and about sleep and dementia in Hindi, which was well received. We were advised by staff that many of the older people spoke multiple languages and that it would still be useful to translate into Hindi. MM and AB divided DREAMS-START sessions between them, independently translating half into Hindi, then swapping sessions to complete blinded back translation, discussing and resolving disagreements. They produced a Hindi script version and a Hindi-in-English phonetic version for individuals who may speak but not read Hindi, thus increasing accessibility. They agreed on phonetic spelling of terms that did not have a literal or appropriate Hindi translation in consultation with NM, who is a Hindi-speaking dementia academic. They expanded or changed phrases to fit Hindi language conventions and developed a glossary of original terms with their phonetic or Hindi script translation. Translations were reviewed by two female British Pakistani qualified social workers working in a South Asian community centre, one attendee with early dementia, and one volunteer staff member, all fluent in Hindi. This iterative internal and external review process followed other language adaption studies [30,31,32]. We consulted Hindi-speaking Patient and Public Involvement (PPI) members, who advised retaining the term ‘carer’ phonetically rather than its Hindi equivalent.

## 3. Results

### 3.1. Participant Characteristics

From September 2022 to July 2023, we identified twenty-three South Asian family carers. Of these, seven declined (due to insufficient time, death of person with dementia, or travelling and family commitments) and five were unreachable. We interviewed eleven carers, of whom seven had received DREAMS-START. MM familiarized the four remaining carers with the intervention (Figure 2).

Nine interviews were in English, one with a Bengali/Sylheti interpreter, and one in Hindi (see Table 1 for demographics). Seven (64%) carers who were the child/child-in-law lived in multigenerational households. In the two instances where the spouse was the carer for the person with dementia, the couple lived alone.

### 3.2. Qualitative Findings

Findings are organized into two overarching themes. The first considers the impact of dementia-related sleep disturbance in South Asian families, and the second explores potential cultural and linguistic adaptation of DREAMS-START.


**
*Theme 1: The impact of dementia-related sleep disturbance in South Asian families*
**

*Sub-theme 1: Sleep disturbances in South Asian multigenerational households*


Carers in multigenerational households described how their relative’s sleep disturbances impacted the entire family. They highlighted that being disturbed at night left each generation feeling tired, affecting their work, school, and caring responsibilities. Carers highlighted how relationships became strained because of pervasive exhaustion:

*We found that she was coming out of her room more and more and going down the stairs […] someone was always awake or going after her […] We had our door open, we had mum’s door open, and the kids would be up, so they’d listen out for mum, they might put her back into bed […] I mean there were times when none of us got any sleep*.(Daughter-in-law)

*I was getting very depressed myself. My whole family was suffering. I had a teenage daughter, she was feeling neglected […] you know, all the energy was going*.(Daughter2)

Carers felt that creating an environment conducive to sleep and establishing a routine could be a challenge in a multigenerational household.

*We think her [relative with dementia] sleep has been affected, I mean, you need to know that we live on one floor… It is a four-bedroom flat… The problem is every bedroom door, […] is almost next to each other… If you touch one door, the person sleeping in the other room… will be affected. I think this is how it all started and then slowly it only got worse*.(Son2)


*Sub-theme 2: The impact of family relationships and cultural expectations on sleep management*


The altered dynamic between the person with dementia and family carer, especially when their child is the carer, could impact the management of sleep disturbances.

*The fact that I’m saying, going against my Mum, right. So, for a child to start caring for the mum… a role reversal, it’s not gone down well. It’s hard for me being the son to tell my Mum or suggest things*.(Son4)

One daughter-in-law explained how their relative with dementia viewed their son as their older brother. Given the respect for older brothers in South Asian cultures, the person with dementia became more receptive to their son’s help in managing sleep problems.

*My husband would then stay in her room and say Mum go to sleep, and I think she saw him an authoritative figure type of thing. She saw him as a brother, she forgot that he was her son, and she’d call him “bhaiyya”, meaning brother […] she would say yes bhaiyya, yes bhaiyya, I’ll do whatever you say Bhaiyya*.(Daughter-in-law)

Carers described cultural expectations and family pressure to manage sleep issues for their relative with dementia without outside help, despite increasing care needs, typically carried by the woman in the family.

*If it’s a woman [family carer], they usually don’t like carers coming in, they expect the families to help out—the daughter-in-law or the women in the house help out. So, … if things get really bad… I know some Asian families where they just struggle*.(Daughter2)

One carer felt that sharing anything at all about having received the DREAMS-START intervention with anyone beyond the immediate family should be avoided, potentially isolating family members.

*Not outside people… So, blood relatives are fine, you know, brothers, his brothers, your sisters, you know, blood relatives know it, and that’s completely fine. Community people don’t need to know*.(Son2)

Carers suggested that any sleep intervention aimed at family carers within the South Asian community would need to acknowledge cultural barriers to accessing help and support carers in asking for help from people inside or outside the immediate family.


*I think understanding that… having a component, addressing that issue of how women in the South Asian community, that they’re seen as being carers anyway and that how do you help them understand that, that they can ask for support and help?*
(Daughter4)

Carers felt it was common among South Asian families to progressively limit the person with dementia’s attendance at family gatherings or events outside of the home. One carer tied this to the challenges of getting their relative out of the house, alongside the wider family and community not understanding certain behaviours resulting from dementia. This is significant for DREAMS-START, which promotes increasing daytime activity to promote nighttime sleep.

*With the Asian community when say somebody has dementia, they don’t take them out as much […] so maybe encouragement of having something more outside the house as well, different activities […] I don’t know how people would be encouraged to go outside, because I know lot of people with an Asian background, when somebody’s got dementia, they’re usually in the house*.(Daughter2)


*Sub-theme 3: Working together to address sleep disturbance*


Carers suggested that extended family members be included or made aware of any intervention aimed at improving a relative’s sleep to ensure its effectiveness. Living and working together to resolve the sleep issues was a potential strength. Carers highlighted that this led to a shared sense of responsibility, but also to consistent support and shared understanding.

*I’d be saying to the family, let’s work together … you know, if there was a particularly noisy family member, saying you need to keep the noise down because grandmother needs to sleep or. So yeah, I think, it would be essential to pass on the information that you were getting onto other family members as well*.(Daughter1)

Shared responsibility helped carers to create the best possible environment for their relative. One live-out carer discussed how their brother’s house was better for their mother in terms of attending to her safety at night.

*It would be completely different if this was just me on my own caring, and my brother wasn’t around. I would have had to change my life completely […] we’d have to sort of reorganize the house and make sure the downstairs is secure and…for the showering and… downstairs and… it would have a big impact for… we were just lucky the way we were able to kind of organize things*.(Daughter1)


**
*Theme 2: Considerations for cultural adaptation of DREAMS-START*
**

*Sub-theme 1: Linguistic adaptation of DREAMS-START*


When discussing widening access to DREAMS-START amongst UK-based South Asian carers, participants reflected on levels of English proficiency amongst carers within the community, which may impact upon their understanding of content and engagement with the intervention.


*It may help those where English isn’t their first language. Because you lose certain, certain nuances, don’t you?*
(Daughter3)

In one case, a family carer reported disengaging completely from the intervention because of difficulty in understanding English, highlighting that the intervention, even when delivered and explained by a facilitator, was challenging:

*language is the main thing right. There are many things that we don’t understand as well in English as we would in our own language. There’s that difference, of course. Even for the sessions, I didn’t find any benefit […] I understood half, I didn’t understand half… it’s a waste of time*.(Wife2)

Carers highlighted that it was helpful to have translations alongside English versions, as this would increase the likelihood of shared understanding across the family:

*it’s also best to give them the English version as well. So, you give them both. So at least if you can’t understand yourself in that Bengali language, you can ask your son or your daughter… and they can explain it to you*.(Son2)

Carers had a mixed response to interpreters. Some felt that interpreters would enable clearer understanding of the content being delivered and capture “nuances” that may otherwise be lost. Other carers felt that an interpreter makes sessions longer and might be intrusive, given the sensitive topics discussed and support needs of the carer.


*Sub-theme 2: Cultural competence of DREAMS-START facilitators*


Some carers thought facilitators being of the same cultural background would put the family carer at ease, allow the carer to discuss specific cultural practices, and have a shared understanding of South Asian culture. Having a facilitator who was culturally competent, curious, and able to respectfully put the DREAMS-START participants at ease was seen as equally valuable.

*It can work to the person’s advantage, if the person is also from the same culture or some similar culture…. Because there’s this kind of shared understanding of how families operate […] I don’t know whether the family would feel embarrassed to say to someone of the different culture. But […] it’s the attitude of the person doing the intervention to just be able to [unclear] and not be judgemental… and accepting of different ways of living*.(Daughter1)

Carers explained that what was important was that the facilitator could communicate effectively and had an understanding of South Asian cultural practices and family cultures.

*As long as they can communicate, like, I’m from an Indian background, I had a White male speaking to me, it didn’t matter that we had different cultures; it mattered that he could communicate well to me and understand my issues*.(Daughter2)


*Sub-theme 3: Potential adaptations of DREAMS-START content*


Carers reflected on whether the peripheral components of the intervention, such as suggested activities or ways to engage people with dementia during the day, fit with their experiences, highlighting how suggested hobbies or interests may not be familiar—although no suggestions are expected to be for everyone. It was suggested that in a translated version of the manual the content would feel more relevant to carers as:

*“it would probably reflect more the sorts of things that people from South Asian families might say…”*.(Daughter1)

*Most of [DREAMS START manuals] use examples about English people, so to speak. And if part of it is slightly changed to the culture of the individual, then that would be helpful*.(Son3)

During the interviews, carers made specific suggestions around how peripheral components could be culturally adapted including connecting the content to religious and spiritual beliefs about caring, making lifestyle and environmental changes that account for religious and cultural dress and head covering, and considering prayer in relation to routine and daily activity which may impact on sleep.

## 4. Discussion

### 4.1. Main Findings

This is to our knowledge the first study of UK-based South Asian family carers’ experience of sleep problems in dementia [19,20]. These findings will inform cultural adaptation of DREAMS-START [12] for the South Asian community, following translation into Hindi.

South Asian families’ multiple caring responsibilities, multigenerational living, cultural expectations of families (especially women delivering care and progressive limitation of going out), and collaborative decision-making with extended family, all impacted on caring for someone with dementia and disturbed sleep. This aligns but adds to previous literature on multigenerational households and shared caring responsibilities among South Asian communities [22], though there may be a gradual reduction in multigenerational households in South Asian communities due to lifestyle and work changes [33]. While families in general prefer care at home, other studies have identified more networks of care, a higher perceived ‘duty to care’, and family members’ apprehensions about their relative with dementia moving to a care home [34]. Previous studies have reported less positive views of the wider family network, which does not always equate to additional support for the primary carer [35] and may be a source of financial support but not necessarily in daily caregiving responsibilities [36,37].

Interviews with family carers and consultation with third sector staff highlighted potential cultural adaptations to DREAMS-START and influenced Hindi translation. Carer views on culturally adapting DREAMS-START did not relate to central therapeutic aspects—none of the intervention components were deemed irrelevant to South Asian communities—but to peripheral elements including the use of colloquial everyday language and facilitators’ cultural awareness and sensitivity. These findings align with other work culturally adapting interventions beyond language translation and maintaining intervention fidelity [38]. It was important that vignettes and examples be culturally representative and use relevant terminology [39]. Previous studies have noted that there is no agreed translation for the word ‘dementia’ in most South Asian languages, thus highlighting the need for clear and appropriate explanations while delivering interventions [40,41].

### 4.2. Strengths and Limitations

Translation of the manuals was undertaken by two bilingual researchers with multidisciplinary consultation and review by South Asian clinicians, social workers, and members of the community affected by dementia, which enabled greater accuracy and quality of translated material [42]. Another strength is the diversity of the sample in terms of the carer’s relationship to the person with dementia, caregiving contexts, and employment status. The small sample size and the majority of family carers being from an educated, Indian background, limits the transferability of the findings. As Hindi is not as widely spoken in UK South Asian communities as Urdu, Punjabi, and Bengali, our work to date is potentially limited, and we aim to extend this in our planned future work. We plan to work with UK South Asian communities to plan future adaptations and will be led by these collaborative discussions to ensure that we are choosing languages for translation that reflect the communities with the most need. Translation into Hindi has been a useful process as it has allowed us to refine our processes for translation and adaptation and will also be useful for adaptation for delivery in India, for which we already have existing collaborations.

### 4.3. Recommendations and Future Directions

We will make these changes to peripheral components of the intervention based on findings, ensuring that examples and prompts are flexible enough to account for diverse lived experiences and add discussion about how important it is for sleep not to progressively limit movement and light. DREAMS-START manuals already encourage individuals to share with family members the key learnings and strategies. We will foreground this in facilitator training to ensure that they are encouraging this practice and feel able to manage multiple perspectives in the sessions, if deemed useful and appropriate. Pictures in South Asian language translations will depict South Asian families.

Adapting to language needs and ensuring cultural competence among facilitators were key to culturally appropriate delivery. We will further develop our facilitator training curriculum to enhance cultural sensitivity. This will have a particular focus on how sleep disturbance may impact individuals living with dementia in multigenerational households, considering religious practices and routines in the context of sleep and how to support families to increase activity and social engagement in a flexible and culturally sensitive way. We also plan to explore delivery through third sector and local community services, for example, via bilingual and peer workers, to further widen access in the future [27]. We have made a Hindi language animation which summarizes our intervention and RCT results, which we hope will be a useful tool in raising awareness and widening access to the intervention.

Nevertheless, family members appreciated the purpose of the intervention and attempt to cover several aspects of managing sleep disturbances in dementia. We achieved a diverse sample in our main trial, with 22.3% of dyads recruited from minoritized communities. This suggests that accessing support for sleep may be more socially acceptable than other aspects of support, and this may therefore be a way to accessing individuals who may otherwise be caring in isolation. As such, we will also highlight the components of our intervention which discuss connecting with wider support structures to address sleep disturbances and beyond.

## 5. Conclusions

South Asian families may have varying experiences of sleep disturbances in dementia due to multigenerational family structures, familism, practices intending to protect the person with dementia but may limit outdoor activity and exposure to light, and less access to support, which should be considered during consultation and intervention delivery. Identifying any potential language barriers prior to delivery is important. Cultural adaptation of the intervention in terms of linguistic translation and cultural competence in delivery was perceived to increase relevance, with potential to increase access within the wider community. Further work is needed to explore differences between South Asian sub-groups, subjective cultural experiences (e.g., migration, religion), and pilot the delivery of the translated and culturally adapted intervention to ascertain acceptability.

## Figures and Tables

**Figure 1 geriatrics-10-00121-f001:**
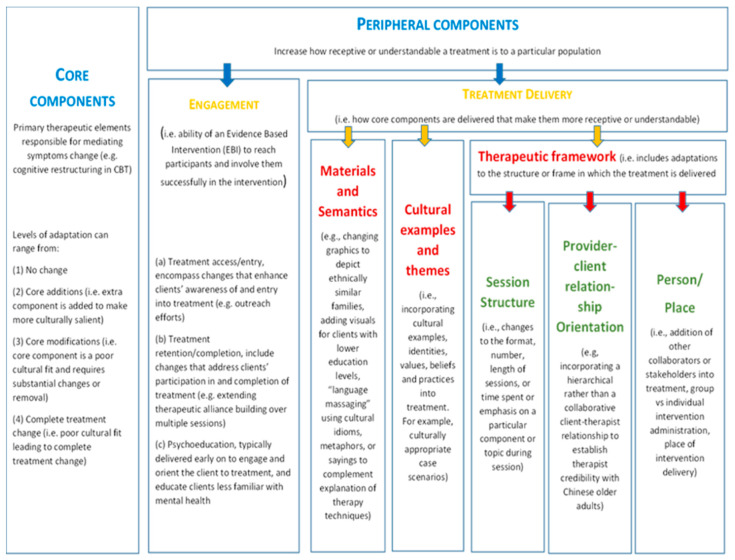
CTAF framework for cultural adaptation.

**Figure 2 geriatrics-10-00121-f002:**
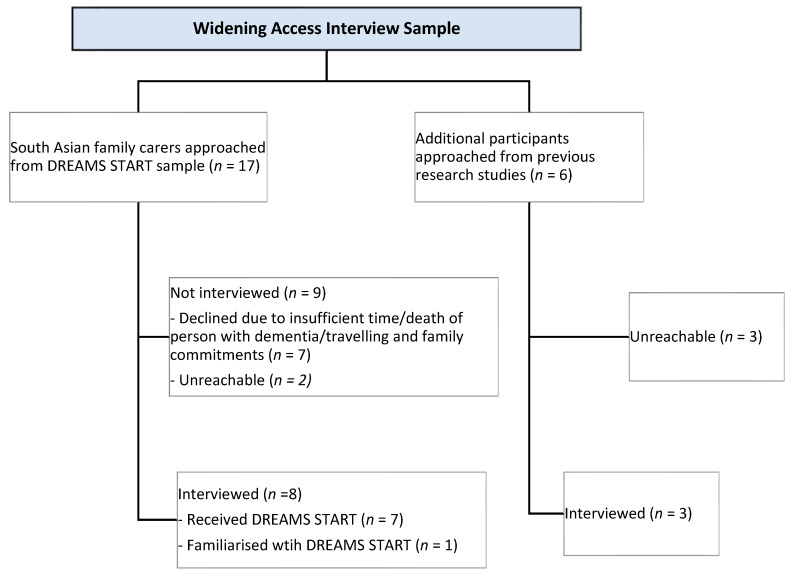
Flowchart showing sample recruited.

**Table 1 geriatrics-10-00121-t001:** Demographic characteristics of family carers interviewed.

Sample Characteristic	Category	n (%) or Mean (SD)
Age		58.8 (10.64)
Sex	Male	4 (36%)
	Female	7 (64%)
First language	English	9 (82%)
	Bengali/Sylheti	1 (9%)
	Punjabi	1(9%)
Ethnicity	Asian/British Asian (Indian)	9 (82%)
	Asian/British Asian (Bangladeshi)	1 (9%)
	Asian/British Asian (Sri Lankan)	1 (9%)
Relationship to person with dementia	Spouse	2 (18%)
	Child	8 (73%)
	Daughter-in-law	1 (9%)
Carer cohabiting with relative when actively caring at home	Yes	8 (73%)
Caregiving status at the time of interview	Caring for relative at home	6 (55%)
	Relative in care home	3 (27%)
	Ex-carer (Relative died)	2 (18%)
Multigenerational household	Yes	7 (64%)
Education	High-school diploma	3 (27%)
	Undergraduate degree	5 (45%)
	Postgraduate degree	3 (27%)
Employment	Full-time	3 (27%)
	Part-time	3 (27%)
	Unemployed	4 (36%)
	Retired	1 (9%)

## Data Availability

Anonymised data will be available on request from the corresponding author.

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
