# Peer review of "Sleep Disturbances and Dementia in the UK South Asian Community: A Qualitative Study to Inform Future Adaptation of the DREAMS-START Intervention"

_geriatrics, 2025, doi:10.3390/geriatrics10050121_

Round 1

Reviewer 1 Report

Comments and Suggestions for Authors

It is commendable that the manuscript explicitly highlights the need to culturally adapt the DREAMS-START intervention for the UK South Asian community, demonstrating clear practical value by directly linking qualitative insights to the adaptation of this established evidence-based programme. The use of the CTAF framework to guide the cultural adaptation process is also a notable strength.

There are a few points for the authors to consider.

  • While Hindi is a widely spoken South Asian language, the manuscript would benefit from a clear rationale for selecting Hindi specifically, given the UK South Asian community’s linguistic diversity, including significant Punjabi, Urdu, Bengali, and Gujarati speakers. It is unclear whether PPI contributors or community representatives were consulted about preferred languages? Clarifying whether the choice was informed by PPI input, local demographic data, or practical feasibility — and noting if other languages were considered or planned for future adaptation — would strengthen the justification and demonstrate a commitment to wider cultural inclusivity.
  • In Table 1, the categories “carer living with relative” and “caregiving context” are unclear — it would help to clarify whether these refer to the carer’s specific role and responsibilities in supporting the individual with dementia?

Author Response

Reviewer 1:

It is commendable that the manuscript explicitly highlights the need to culturally adapt the DREAMS-START intervention for the UK South Asian community, demonstrating clear practical value by directly linking qualitative insights to the adaptation of this established evidence-based programme. The use of the CTAF framework to guide the cultural adaptation process is also a notable strength.

Thank you for this feedback.

There are a few points for the authors to consider.

While Hindi is a widely spoken South Asian language, the manuscript would benefit from a clear rationale for selecting Hindi specifically, given the UK South Asian community’s linguistic diversity, including significant Punjabi, Urdu, Bengali, and Gujarati speakers. It is unclear whether PPI contributors or community representatives were consulted about preferred languages? Clarifying whether the choice was informed by PPI input, local demographic data, or practical feasibility — and noting if other languages were considered or planned for future adaptation — would strengthen the justification and demonstrate a commitment to wider cultural inclusivity.

As noted there is wide linguistic diversity amongst UK South Asian communities and we have added further detail to our methods and extended comment in our limitations to address this. This was a relatively small study which was our first steps towards cultural adaptation and as such our choice of language was pragmatic based on the languages spoken within our research team which enabled us to conduct the translation with limited resource. We consulted with staff and members of a South Asian community centre as well as our team member Naaheed Mukadam – who has written extensively on dementia and ethnicity, especially in the context of the UK South Asian Communities. We have added to our methods:

“Whilst Hindi is less commonly spoken in the UK than other South Asian languages such as Urdu, Punjabi and Bengali, we made a pragmatic decision within the limitations of this small-scale study to translate into Hindi as MM and AB were native Hindi speakers and were able to lead on the translation process. We consulted with a leaders in a South Asian community centre who highlighted, presenting to a group of older people on sleep and dementia in Hindi, which was well received, we and were advised by staff that many of the older people spoke multiple languages and that it would still be useful to translate into Hindi.”

We have added to our discussion of limitations:

“As Hindi is not as widely spoken in UK South Asian communities as Urdu, Punjabi and Bengali our work to date is potentially limited and we aim to extend this in our planned future work. We plan to work with UK South Asian communities to plan future adaptations and will be led by these collaborative discussions to ensure that we are choosing languages for translation that reflect the communities with the most need. Translation into Hindi has been a useful process as it has allowed us to refine our processes for translation and adaptation and will also be useful for adaptation for delivery in India, for which we already have existing collaborations.”

In Table 1, the categories “carer living with relative” and “caregiving context” are unclear — it would help to clarify whether these refer to the carer’s specific role and responsibilities in supporting the individual with dementia?

Apologies for this lack of clarity – we have relabelled these categories to make clearer. “Carer living with relative” is now labelled “Carer cohabiting with relative when actively caring at home” and “Caregiving context” is now labelled “Caregiving status at the time of interview”.

Reviewer 2 Report

Comments and Suggestions for Authors

Dear Authors

Your submitted manuscript title "Sleep Disturbances and Dementia in the UK South Asian Community: A Qualitative Study to Inform Future Adaptation of the DREAMS-START Intervention " conducts qualitative survey analysis in the selected population and concludes future adaptation of the Dream-start intervention.

Abstract: The Authors have written well.

Introduction: This section is also well-written and covers all aspects of the objective.

Methodology: It's appropriate for this study, but I recommend including more samples or increasing the sample size if possible. 

Result: The results are interesting,

Conclusion: Ok 

References: It's ok

Author Response

Reviewer 2:

Your submitted manuscript title "Sleep Disturbances and Dementia in the UK South Asian Community: A Qualitative Study to Inform Future Adaptation of the DREAMS-START Intervention " conducts qualitative survey analysis in the selected population and concludes future adaptation of the Dream-start intervention.

Abstract: The Authors have written well.

Thank you.

Introduction: This section is also well-written and covers all aspects of the objective.

Thank you.

Methodology: It's appropriate for this study, but I recommend including more samples or increasing the sample size if possible.

As this study has now been completed, we are not in a position to increase our sample size of 11 at this point. We believe that this study has merit and as an exploratory, qualitative study we feel that even with this number of interviews our findings are interesting and add to what is known and understood about sleep and dementia in the UK South Asian Community. As is the case with qualitative studies we were not aiming for generalisability but for the development of transferable information which contributes to wider understandings of a topic. And we acknowledge in our limitations section that the small sample size limits this transferability. In line with existing guidance (1) we continued our interviews until we felt that thematic saturation was achieved and no new themes were being identified in our analysis.

We have added to our methods: “We ceased interviews at thematic saturation, at the point that the researcher coding an interview identified no new codes and when the authors’ reflections on additional inter-views resulted in no further emergent themes.”

We do plan to do future adaptation work building on this study and will at this point conduct a larger scale feasibility study of the adapted intervention.

Result: The results are interesting,

Thank you

Conclusion: Ok

Thank you

References: It’s ok

Thank you

Reviewer 3 Report

Comments and Suggestions for Authors

Although this manuscript is well-written, the study is clearly improvable. There are few subjects in the study. You have to introduce more subjects in the study (only 8 interviewed). 

Reconsider this fact. 

Author Response

Although this manuscript is well-written, the study is clearly improvable. There are few subjects in the study. You have to introduce more subjects in the study (only 8 interviewed).

Reconsider this fact.

 As this study has now been completed, we are not in a position to increase our sample size of 11 at this point. We believe that this study has merit and as an exploratory, qualitative study we feel that even with this number of interviews our findings are interesting and add to what is known and understood about sleep and dementia in the UK South Asian Community. As is the case with qualitative studies we were not aiming for generalisability but for the development of transferable information which contributes to wider understandings of a topic. And we acknowledge in our limitations section that the small sample size limits this transferability. In line with existing guidance (1) we continued our interviews until we felt that thematic saturation was achieved and no new themes were being identified in our analysis.

We have added to our methods: “We ceased interviews at thematic saturation, at the point that the researcher coding an interview identified no new codes and when the authors’ reflections on additional interviews resulted in no further emergent themes.”

Additionally, many qualitative studies have small numbers of participants (e.g.<15) and it has been established (1) that thematic saturation can be achieved at 12 interviews with major themes emerging even after 6 interviews.

  1. Guest, G., Bunce, A., & Johnson, L. (2006). How many interviews are enough? An experiment with data saturation and variability. Field methods, 18(1), 59-82.

Round 2

Reviewer 3 Report

Comments and Suggestions for Authors

Although qualitative studies aren't the best due to their low number of participants, I appreciate that you clearly explained this in your comment between lines 111 and 114. However, quantitative studies should be considered in the future, as they are more accurate in obtaining potentially statistically significant results.